# A SWOT Analysis for Offshore Wind Energy Assessment Using Remote-Sensing Potential

**Meysam Majidi Nezhad [1], Riyaaz Uddien Shaik [2], Azim Heydari [1], Armin Razmjoo [3], Niyazi Arslan [4] and Davide Astiaso Garcia [5,\*]**

[1]  Department of Astronautics, Electrical and Energy Engineering (DIAEE), Sapienza University of Rome, 00184 Roma, Italy; meysam.majidinezhad@uniroma1.it (M.M.N.); azim.heydari@uniroma1.it (A.H.)
[2]  School of Aerospace Engineering, Sapienza University of Rome, 00184 Rome, Italy; riyaaz.shaik@uniroma1.it
[3]  Escola Tècnica Superior d'Enginyeria Industrial de Barcelona (ETSEIB), Universitat Politècnica de Catalunya (UPC), 08028 Barcelona, Spain; arminupc1983@gmail.com
[4]  Department of Geomatics Engineering, Cukurova University, Ceyhan Campus, 01950 Ceyhan, Turkey; niyazi.arslan@gmail.com
[5]  Department of Planning, Design, and Technology of Architecture (DPDTA), Sapienza University of Rome, 00197 Rome, Italy
\*  Correspondence: davide.astiasogarcia@uniroma1.it; Tel.: +39-06-4991-9174

**Abstract:** The elaboration of a methodology for accurately assessing the potentialities of blue renewable energy sources is a key challenge among the current energy sustainability strategies all over the world. Consequentially, many researchers are currently working to improve the accuracy of marine renewable assessment methods. Nowadays, remote sensing (RSs) satellites are used to observe the environment in many fields and applications. These could also be used to identify regions of interest for future energy converter installations and to accurately identify areas with interesting potentials. Therefore, researchers can dramatically reduce the possibility of significant error. In this paper, a comprehensive SWOT (strengths, weaknesses, opportunities and threats) analysis is elaborated to assess RS satellite potentialities for offshore wind (OW) estimation. Sicily and Sardinia—the two biggest Italian islands with the highest potential for offshore wind energy generation—were selected as pilot areas. Since there is a lack of measuring instruments, such as cup anemometers and buoys in these areas (mainly due to their high economic costs), an accurate analysis was carried out to assess the marine energy potential from offshore wind. Since there are only limited options for further expanding the measurement over large areas, the use of satellites makes it easier to overcome this limitation. Undoubtedly, with the advent of new technologies for measuring renewable energy sources (RESs), there could be a significant energy transition in this area that requires a proper orientation of plans to examine the factors influencing these new technologies that can negatively affect most of the available potential. Satellite technology for identifying suitable areas of wind power plants could be a powerful tool that is constantly increasing in its applications but requires good planning to apply it in various projects. Proper planning is only possible with a better understanding of satellite capabilities and different methods for measuring available wind resources. To this end, a better understanding in interdisciplinary fields with the exchange of updated information between different sectors of development, such as universities and companies, will be most effective. In this context, by reviewing the available satellite technologies, the ability of this tool to measure the marine renewable energies (MREs) sector in large and small areas is considered. Secondly, an attempt is made to identify the strengths and weaknesses of using these types of tools and techniques that can help in various projects. Lastly, specific scenarios related to the application of such systems in existing and new developments are reviewed and discussed.

**Keywords:** marine renewables; remote sensing; offshore wind; SWOT(strengths, weaknesses, opportunities and threats) analysis

## 1. Introduction

Human societies are currently facing global warming, and national governments are working to find solutions against the climate crisis, by promoting the installation of renewable energy sources (RESs) to replace fossil fuels. This energy transition could avoid the emissions of GHG (greenhouse gases) [1,2] and reduce the use of carbon fuel reserves [3]—minimizing at the same time geopolitical conflicts to access oil and gas sources [4]. Additionally, this green energy transition is creating new challenges in different sectors and consequentially new job opportunities in economic, technological, and environmental topics [5], together with other kinds of economic and social benefits [6]. With the aim of reducing carbon exploitation, many power energy converters have been designed and installed, considering the specific contexts of each analyzed area. Depending on the sources of power extracted from seas and oceans, two different categories should be defined: "blue energy" (BE) and "marine renewable energies" (MREs). Blue energy, such as salinity gradients, thermal differences and MREs including sea surface waves, sea current, tides, wind, geothermal and solar. Assessing the renewable energy sources (RESs) availability is important in developing short and long-term planning [7,8]. In this regard, wind energy could be one of the safest and most reliable sources of renewable energy [9]. To use these renewable sources, many aspects must be examined. Among these, it is fundamental to assess the exact amount of power for each type of energy converter. This has given more attention to the development of new offshore solutions, such as wind turbines with larger rotors, deep water foundation and floating platforms [10]. Northern and central European countries (ECs) have a long history of designing, developing and installing offshore wind farms [11], since the installation of first prototypes of bottom fixed and floating offshore wind farm in Baltic Sea and Scotland pilot park [12]. Nearly 90% of the world's MREs are in Europe. However, the proportion of the Mediterranean in the use of this vast resource is extremely low [13].

Among those with a feasible amount of wind energy source, the best suitable sites for offshore wind should be selected mainly according to the optimal combination of water depth and distance to the shore [14]. Water depth is a key factor in better understanding the dynamics of the marine environment, in predicting tides, currents and waves and planning offshore facilities and infrastructure such as wind turbines. Selecting suitable places with the optimal combination of water depth and distance to the beach is a complex task and should be carefully examined. The coefficient of water depth in the desired area is the basic parameter in the type of wind farm installation, mainly due to the maintenance and installation cost increase. In addition, the distance from the desired point to the power grid is very important, which increases due to the greater distance, which leads to more use of cables and batteries. Floating platforms could help in this regard which is the current trend to move considering deeper waters. Furthermore, there are more crucial factors/limitations influencing offshore wind (OW) applications, such as ships sea routes for marine transportation, migrating birds, economic activities (e.g., fisheries areas), environmental constraints (marine protected areas) and landscaping view. To combine economic sustainability and public acceptance, the concept of floating marine turbines operating away from coastal areas has also begun and expanding in the Mediterranean Sea [14]. Together with the growth of onshore wind farms installations, there is an expectation for significantly increasing the new wind farms in offshore marine areas, mainly thanks to floating platform technologies [15]. The dramatic growth of new technologies has led to an immediate revolution in the use of new offshore wind farms, statistically, with the wind farm sector in continental Europe showing an annual growth of 101% in 2017 [16].

In general, the installation capacity of wind farms in 2019 for European countries (ECs) is 21.1 GW and there is an expectation that in 2020, total OW energy production could reach 25 GWh, and by 2030,

it could reach to 70 GW in offshore installation capacity [17,18]. Totally, ECs have 4149 grid-connected wind turbines and 81 offshore wind farms installed, which are used only in 10 countries of northern Europe. According to data from 2017, about 50% of offshore wind farms in continental Europe were installed in UK (which is 53% of the net 3.15 GWh of installed capacity in Europe). By 2024, Europe's total installed capacity is expected to reach 29.8 GW, expanding at an annual growth rate of 12% [19]. On the other hand, despite all these considerations, there are still no OW farms installed in the Mediterranean Sea, mainly for the water depth that usually does not allow the installation of bottom fixed wind turbines; anyway there are attractive hot spots for future developments of OW in the Mediterranean [20].

The first step in installing wind farms is to evaluate wind sources in focused areas or hot spots. Traditional methods, such as cub anemometers, are frequently used to measure wind sources, by installing calibrated calipers on tall masts. It should be noted that the height of the wind gauges is directly related to the height of the installed wind turbines. Due to the significant growth of technology, the height of wind turbines is constantly increasing. Consequentially, taller masts are needed, increasing the installation costs and the operation and maintenance efforts. In addition to all the above consideration, it should be noted that natural and human obstacles are the most important factors for the installation of anemometers. The fewer natural and human factors in the area, the fewer wind gauges are needed. However, if there are natural and human obstacles such as cities, more wind gauges are needed [21]. On the other hand, due to the significant development of on-site remote sensing sector, LiDAR (light detection and ranging) and SODAR (sonic detection and ranging) tools could be applied. Goit et al. [22], explained the though reconstruction from LiDAR-measured radial wind speed to wind speed vector is a challenge, LiDAR-based wind speed measurements are undergoing a significant increase in interest for wind energy application. Here, the study employed the scanning of Doppler LiDAR for assessment and comparison. First, the evaluation of the effect of carrier-to-noise-ratio (CNR) and data available on the quality of scanning LiDAR measurements was done. Then, it was proposed to reconstruct the wind fields from plan-position indicator (PPI) and range height indicator (RHI) scans of LiDAR-measured line of sight velocities. It was observed that an internal boundary layer with strong shear could be developed from the coastline. Lastly, PPI scan was involved to measure the flow field around a wind turbine and validate wake models. Chaurasiya et al. [23] investigated how to increase the confidence of RS technique to compute Weibull parameters at higher heights for assessment of wind energy resource. It is known that RS techniques are gaining attention worldwide for their comprehensive assessment of wind source in flat, complex, and mountainous terrain. The 10 min average time series wind speed data for the period of one month (in September 2014) were recorded simultaneously at 80 m and 100 m using the cup anemometer installed in the proximity of a 120 m meteorological mast, second wind triton SODAR (sound detection and ranging) and continuous wave wind LiDAR (light detection and ranging). Nine different methods have been implemented to analyze and obtain Weibull parameters on the data obtained from the measurements. Totally, there is an expectation that the outcome of this study could encourage the deployment of remote sensing techniques.

However, this equipment is very expensive and needs to be installed in a study area for more than one year to get enough data; on the other hand, high maintenance and repair costs are required [24,25]. Consequentially, it is very important to develop new methods that can help to identify suitable areas faster and economically. Satellites are the only tool that can measure the average, minimum and maximum wind speed in a study area (hot spot areas) in the shortest possible time. It should be noted that the reasons for the popularity of these data in the research and academic communities can be mentioned as follows: (i) This data are generally available for free (open access), (ii) They can cover a period of more than 40 years. Due to the fact that it is not possible to install ground wind gauges (such as, cub anemometers) in different areas, due to its high cost, satellites are the only device that can cover areas for more than a year, which is an important factor in assessing wind resources [21]. Satellite technologies for observing, reporting, and evaluating RES potential have led to a revolution



in the installation of energy converters in new locations that have not previously been considered. In addition, due to the increase in surveys to identify industrial wind at an altitude of more than 100 m, various atlases were generated as an outcome. It takes long time to install anemometers on-site to measure industrial winds at higher altitudes which can be done with satellite data. However, to bring out the best options and strategies, all aspects of SWOT (strengths, weaknesses, opportunities and strengths, weaknesses, opportunities and threats) must be considered. However, all aspects of SWOT need to be considered to point out the best options and strategies [26]. A SWOT analysis can be used to achieve this goal. This type of analysis, derived from an interdisciplinary approach, can identify barriers and factors influencing the development of marine renewables.

Pisacane et al. [27] explained the current prospects for the exploitation of power plants in the Mediterranean Sea, outlining and discussing challenges, opportunities, and limitations for the deployment of power converters. It was stated that blue energy conversion technologies are now ready to be fully deployed in the device farms. Goffetti et al. [26] described a SWOT analysis of strategic plans for marine renewable energy technologies to minimize and maximize inefficiencies and energy production. SWOT analyses have been used for the navigation of renewable energy technologies and identifying key or hot spots points in various sectors including social, economic, legal, technological, and environmental. Nikolaidis et al. [28] investigated the status of marine renewable energy potential in the Mediterranean Sea and especially around the island of Cyprus. According to their study, OW energy in the Mediterranean Sea is the prominent outlet followed by marine biomasses. On the other hand, they explained that the main physical parameter for developing marine renewable energy projects around that islands is bathymetry. Azzellino et al. [29], using a spatial planning approach, explained the feasibility of installation by choosing a best location for wind turbine generators (WTGs) and wave energy converters (WECs). They proposed a quantitative spatial approach to identify potential sites of interest for the development of marine renewables with an effective perspective, by considering and minimizing potential environmental impacts. The obtained results showed that the Tyrrhenian coastline surrounding the island of Elba, the Northern and Western Sardinian coasts and the Adriatic Sea and Ionian coastal waters, were the most suitable sites for installing marine energy converters. Moreover, further studies about SWOT analysis locations are available.

In the light of the above considerations, the main aim of this research is to develop for the first time a framework on SWOT analysis for investigating the strengths, weaknesses, opportunities and threat of remote sensing (RS) techniques to measure potential power from OW installations. In particular, the present study aimed at elaborating a SWOT analysis for assessing the wind energy potential with RS techniques in the biggest islands of the Mediterranean Sea: Sicily and Sardinia. This study raises a better understanding of how to use RS technology to replace traditional wind measurement tools. The results of the SWOT analysis are expected: (i) to highlight satellites' ability to measure marine renewables; (ii) to identify pros and cons of using these techniques.

### 1.1. Synthetic Aperture Radar (SAR) Satellites

The first European Space Agency (ESA) satellite was launched in July 1991 and called the European remote sensing satellite (ERS-1) [30], which is a C band (5.3 GHz) microwave instrumentation (AMI) satellite [31]. The ESA has recently designed, developed, and launched new satellites so-called Sentinel family providing free data after a simple registration. These satellites could be used to conduct research on various parameters of RESs, for example wind speed, wind direction, wave height, tidal, thermal ocean water and ocean depth measurement. Several SAR satellites and scatterometer, such as QuikSCAT, OSCAT, ASCAT-A, ASCAT-B and Sentinel have already in use for this purpose. Sentinel family which is a group of satellites orbiting around the earth with varying revisiting time for observing land, ocean and atmosphere from space and then for providing us the data free of charge anytime around the year (24/7 and 365 days). One of the most ambitious and world's largest earth observation program in existence today is the Copernicus. Previously known as GMES (global monitoring for environment and security), aiming to tackle environmental challenges with a fleet of autonomous

satellites. Starting from global warming to land use change and the atmosphere, Copernicus gather earth data from space and in the center of it all is its Sentinel family of satellites. Satellites are largely used for different kind of applications [32].

Karagali et al. [33], explained the characterization of the near-surface winds over the northern European shelf seas using satellite data, including the inter- and intra- annual variability for resource assessment purposes. Comparison of mean winds from QuikSCAT with reanalysis fields from the WRF (weather research and forecasting) model and in situ data from FINO-1 offshore research mast was carried out. By this study, the applicability of satellite observations as the means to provide useful information for selecting areas performing higher resolution model runs or for mast installations. It is observed that there were biases ranging mostly between 0.6 and −0.6 ms$^{-1}$ with a standard deviation of 1.8–2.8 ms$^{-1}$. The combined analyses of inter- and intra- annual indices and the wind speed and direction distributions allow the identification of three subdomains with similar intraannual variability. High-resolution satellite SAR wind fields were depicted to observe the local characteristics from the long-term QuikSCAT wind rose distributions. The WRF reanalysis dataset misses seasonal features observed for QuikSCAT and at FINO-1 winds.

Bentamy et al. [34] considered some more specific areas for studying and assessing the offshore wind power potential. To achieve their objective, requirement was given on wind speed and direction with enough spatial and temporal sampling under all weather conditions during day and night. For more than 12 years of remotely sensed consistent data that were retrieved from ASCAT and QuikSCAT scatterometer estimations, were used to find conventional moments associated with wind distribution parameters and then the latter comparable to wind observations from meteorological stations. Further improvisation was carried out by combining in situ and scatterometer wind information. Wind statistical results were used to study the spatial and temporal patterns of the wind power. Also, they depicted the main parameters characterizing wind power potential such as variability, mean, maximum energy, wind speed and intraannual exhibit seasonal features and interannual variability and then found the differences between the wind power estimated for northern and for southern Brittany. Signell et al. [35] detailed spatial structure of jets using the in-situ observations that were carried out on northeast Bora wind events in the Adriatic Sea during the winter. For this, high resolution spaceborne RADARSAT 1 SAR images collected during the active Bora period of the year 2003, created a series of high-resolution maps of 300 m dimension. Along with the previous observations on Bora winds, in this study, it was understood that along the Italian coast, several images show a wide (20–30 km) band of northwesterly winds that abruptly change to northeasterly Bora winds further the offshore. It was concluded by meteorological model that northwesterly winds are consistent with those of a barrier jet forming along Italian Apennine mountain chain.

*1.2. ECMWF Reanalysis*

ECMWF is the European Centre for Medium-Range Weather Forecasts, this organization is advancing numerical weather prediction through global collaboration. The ECMWF reanalysis dataset was chosen for different studies, because of its high temporal and spatial resolution [36]. ECMWF dataset covers a vast timespan starting from 1979 to the present time [37]. ECMWF Reanalysis datasets, including ERA5, ERA-interim, ERA-interim/land, CERA-SAT, CERA-20C, ERA-20CM, ERA-20C. Furthermore, ECMWF data gathers information from the global observation system comprehending different kind of satellites, meteorological, buoys, weather stations and ships.

These limitations are mentioned above, that do not include RS satellite technology and reanalysis datasets [38], such as ECMWF, MERRA, NARR and ERA. A reanalysis dataset is the only source resolved long-term information of spatial wind information at wind turbine height. It provides data and potential assessment of this sector strongly related to the quality form of the meteorological situation [39]. Due to high costs and limitations of the measuring device, including coastal stations, buoys, ships, masts, the use of this kind of dataset can give possibility to understand wind speed with good quality in wind farm site assessment. This step is a very important criterion in possible

alternatives that can be provided by high-resolution reanalysis data. Furthermore, this kind of analysis can reduce the cost of on-site wind measurement [39]. Arun Kumar et al. [40], described the limits of offshore buoys, pinpointing that wind source assessment would be a challenging task. One of the best alternatives for inexpensive data and for filling the data gaps by providing a huge volume of data for extended periods is satellite RS. The assessment of wind source from single scatterometer could lead to inconsistency where there is a requirement of multiple satellite scatterometer. Therefore, four scatterometers viz. OSCAT, ASCAT-A, ASCAT-B and QuikSCAT with long-term in situ wind datasets over the North Indian Ocean were considered. It has been observed that QuikSCAT and OSCAT wind data have lesser bias with the range of 0.15 m/s (2.4%) to 0.83 m/s (15.1%) before adjustments. Linear regression was used for adjustment and the synergetic approach of linear regression adjustments and the combination of scatterometer data have resulted in smaller differences.

## 2. Case Studies

Many islands are not connected to the main grid and still significantly depend on fossil fuels for covering their energy needs. This approach is no more sustainable considering the economic aspects, environmental externalities, high costs in electricity generation and the high amounts of pollutant emissions. Furthermore, such isolation often leads to a high dependency on foreign countries, that could be solved with better solutions for increasing the renewables penetration in small or big islands [41]. Statistically, considering more than 85,000 islands all around the world, approximately 13% of them are inhabited with a population of around 740 million [42]. About 21 million people live on 2050 small islands, each with a population between 1000 and 100,000 inhabitants. Electricity demand in these islands is around 52,690 GWh; anyway, almost half of these islands is in the Pacific Ocean and still have no access to electricity [43].

As regard the Mediterranean Sea, satellite data can be used to measure renewable energy potential for exploiting marine renewables to meet islands energy needs. The exploitation of marine RESs in the Italian islands is crucial to gradually replace their heavy dependence on fossil fuels. On the other hand, Italy has a coastline of more than 8000 km, including 458 small islands with an interesting potential for installing WTGs and WECs [26]. Many of these islands have a higher wind potential compared to coastal areas in the mainland [44]. The highest wave energy potential in Italy is mainly located on the west coast of Sardinia and Sicily [45]; in particular, wave power at the coast is estimated as 10 kW/m in the west coast of Sardinia and 4.5 kW/m in the West Sicilian coastline [46]. Figure 1 shows the location of the two case study areas in the Mediterranean Sea. The red area (West of Sicily) covers three small Italian islands (Favignana, Marettimo and Levanzo) and the green area covers the Southwestern region of Sardinia in same ERA-interim pixel size (6 pixels for each regions).

One important parameter for installing marine energy converters is bathymetry, which is difficult in the Mediterranean Sea and does not allow for the installation up to several hundred meters off the coast due to the high depths [47]. Bathymetry and power potential availability are key parameters to be considered for developing the technical feasibility of designing and installing WTGs and WECs at sea [48]. However, technical, and economic feasibilities related to the use of floating platforms for wave and wind offshore farms are quickly improving.

In this context, wind is considered a very promising source for WTG installation and future WEC installation on the islands, respectively. Hence, various studies were carried out to identify the best places to utilize those energy sources. Even if each RES has limitations on the use and installation of converter devices, they can also use in combination. This combination of energies could be especially applicable to small islands in the Mediterranean, as they have many commonalities such as size, available RESs, weather conditions, population, and environmental constraints [41]. On the other hand, the two most important parameters to consider while identifying the feasible locations are tourist activity and landscaping constraints [49] because tourism is one of the key economic activities [50] in most of the Mediterranean Sea islands, mainly in summertime. The Mediterranean region is one of the most popular tourist destinations in the world and attracts one-third of international tourists.

The number of these tourists is expected to reach 500 million in 2030 [51]. The increase in significant energy demand during the summer season can be covered by these potential renewable energy sources.

Furthermore, many of these islands are included in marine protected areas, where the installation of power converters such as WTGs and WECs can be forbidden for environmental reasons. Generally, many factors and parameters must be specifically considered when suitable areas for wind turbine installation are close to environmental protected areas [52]. With respect to environmental issues, comparatively, fossil fuel power plants pollute more than wind farms. For example, noise caused by rotor blades, could affect the behavior of living species such as dolphins, fish, bats and birds [53]. Industry and researchers are working to reduce the negative effects of WTGs on wildlife by taking preventive measures [6] by choosing the proper location of a wind farm to reduce the bird mortality rate. Many studies were aimed to assess and mitigate environmental impacts of WTGs in marine areas, such as birds and bats and other wildlife species [54].

The installation of OW farm technologies dramatically changes the shape of the offshore realm, which may cause conflicts between communities and developers. Considering this issue, evaluation and analysis of wind sources play a key role in selecting suitable locations for the construction of OW farms. Wind energy assessment, planning and installations [55] must be carried out considering important parameters such as mean wind speed (MWS), wind energy density (WPD) and Weibull parameters [56].

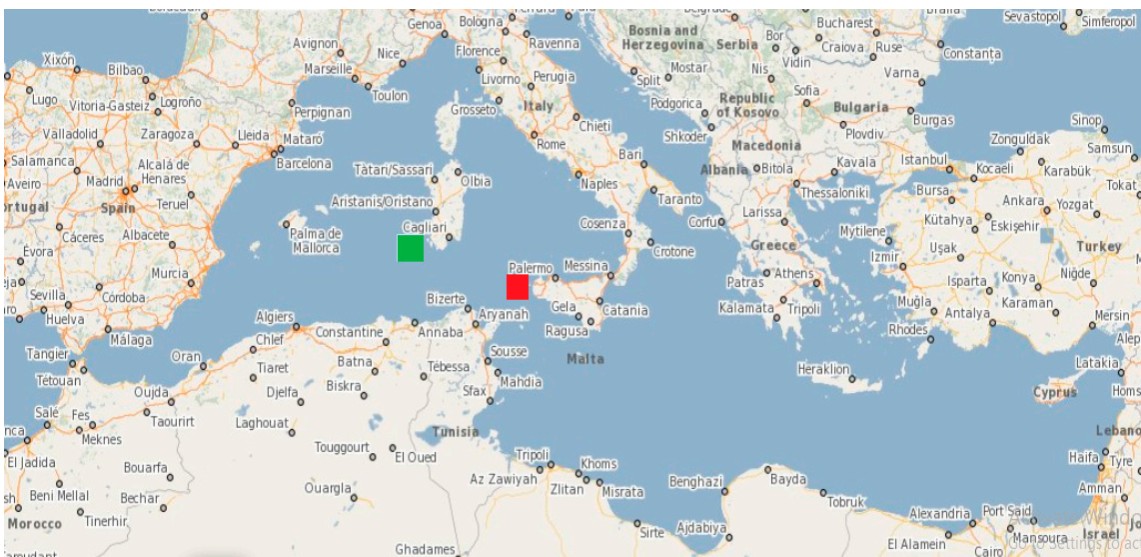

**Figure 1.** Two case studies near the Sicilian and Sardinia islands (with red and green boxes) in the Mediterranean Sea.

## 3. Material and Methods

### 3.1. Satellites and Reanalysis Analysis

The developed method is based on an integrated approach for the preliminary wind speed assessment using Sentinel 1 and reanalysis data by, (i) Sentinel application platform (SNAP) software and (ii) environment for visualizing images (ENVI) software [57]; (i) SNAP stands for sentinel application platform, which is a common architecture for all Sentinel toolboxes, was jointly developed by Brockmann Consulting, SkyWatch and the C-S. This is an ideal software for Earth observation (EO) processing and analysis due to the various technological innovations such as modular rich client platform, extensibility, portability, tiled memory management, generic EO data abstraction and a graph processing framework [58].

(iii) ENVI Software, meaningful information from imagery can be extracted from satellite imagery using this software to make better decisions [59]. This is one of the popular and user-friendly

software in the field of RS, which is mostly used by RS scientists, image analysts and geographical information system (GIS) professionals. This software could be accessed from the desktop, in the cloud and on mobile devices and could also be customized through an API to meet specific project requirements. It uses scientifically demonstrated analytics to deliver expert-level results and also various businesses and organizations preferred ENVI because it has shown easy integration with existing workflows, supported most popular sensors and could easily be customized to meet unique project requirements [59]. ROI (region of interest) tool in ENVI is one of the most used tools in the many applications and have been in from many years and in many processes since the development of its first version called ENVI classic. This tool is used to select the ROI in the satellite image for further analysis or vice versa. Usually, ROI can be selected with geometric shapes like square, polygon, etc., but the drawback of using ROIs is that they are based on image coordinates (number of rows and columns) rather than map coordinates which means they are not easily transferred between images of different sizes or projections. Map coordinate-based vectors (shapefiles and ENVI evf) are more frequently utilized because they are more portable between images and between image processing packages [60]. However, there are still many uses for ROIs and in ENVI there is a new method for their creation too. In this study, preliminary data were obtained by ECMWF reanalysis dataset. For ERA-interim processing, it was carried out in two main steps, namely: (i) analyze era-interim data with GIS software for mapping; (ii) wind speed analysis using the ROI tool.

The first step involves the use of GIS software for mapping wind potential and other parameters [61] in the study area as shown in Figures 2–5. The second step enables the user to analyze the potential of wind energy in a specific area focusing on different zones [62] or hot spots, such as the west part of Sicilian and Sardinia Islands. The ROI tool was used for classification, masking, and operations, also for automatically retrieving information and statistics about a specific area in a larger or smaller area. At this point, after identifying the specific area, all the layers can be merged as one layer to make a time series analysis according to a different time steps and research grope targets (per day, monthly or yearly).

Figures 2–5 show MWS (m/s), mean wind power (MWP), significant wave height (SWH) (m) and sea current speed (SCS) (m/s), analyzed using ECMWF reanalysis dataset for the Mediterranean Sea. As already mentioned, the biggest problem in the design, development, and installation of MRE converters in the Mediterranean Sea is water depth, due to the steep slopes around the shorelines.

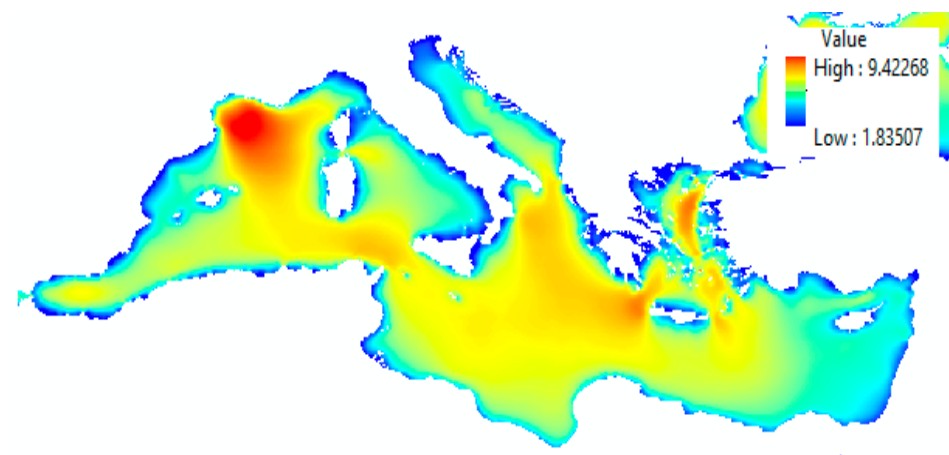

**Figure 2.** Mean wind speed (MWS) (m/s) with 10 m height in the Mediterranean Sea for the years from 2010 to 2017.

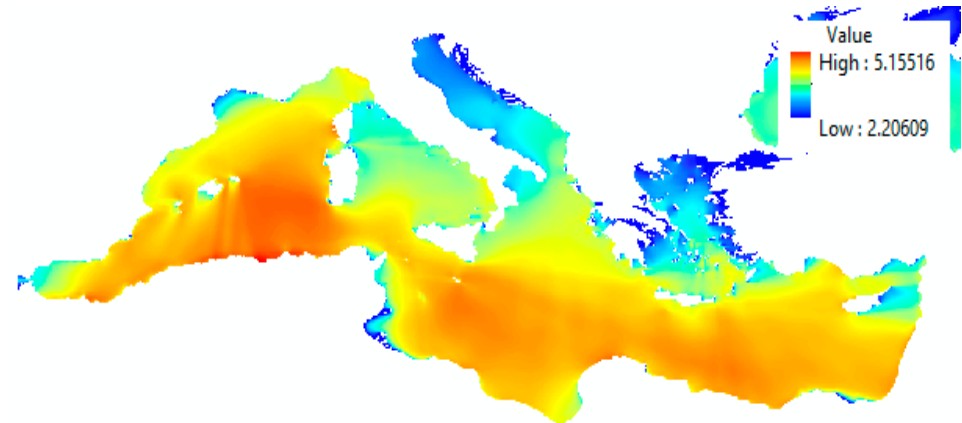

**Figure 3.** Mean wind power (MWP) (kW) with 10 m height in the Mediterranean Sea for the years from 2010 to 2017.

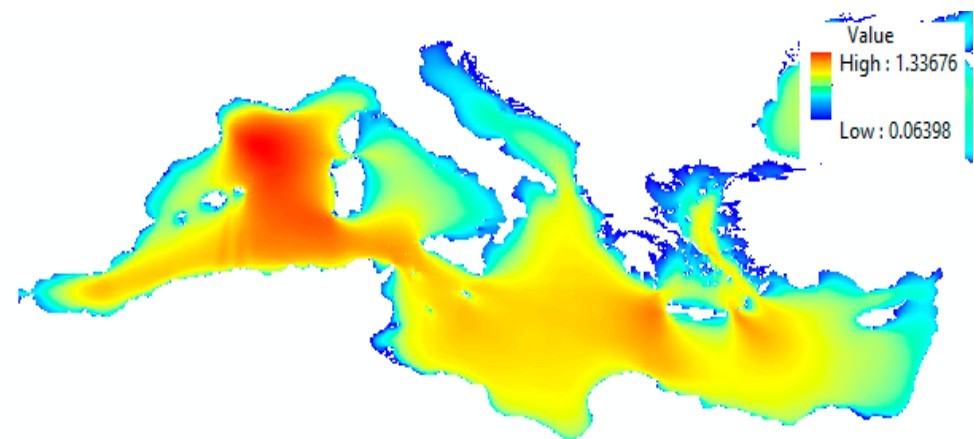

**Figure 4.** Significant wave height (SWH) (m) in the Mediterranean Sea averaged for the years from 2010 to 2017.

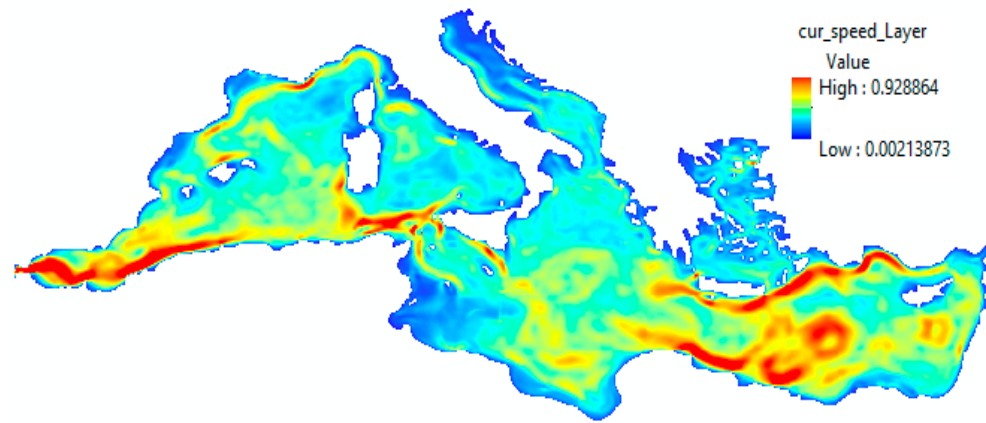

**Figure 5.** Sea current speed (SCS) (maximum speed over layer depth) averaged with 10 m height for the years from 2010 to 2015.

*3.2. SWOT Analysis*

A SWOT analysis was carried out to identify areas of interest for marine renewables installation around large or small islands. This type of analysis is used for small and medium strategic planning [27]. Moreover, some researchers suggested that SWOT analysis can be used as an appropriate tool for

the development of sustainable energy strategy and strategic environmental assessment systems at national level [63].

The purpose of this type of analysis is to design a qualitative process structure that identifies the changes with main strengths, weaknesses, opportunities, and threats to the qualitative structure of the system under consideration. Currently, there are many research articles on the evaluation of renewable energy at different scales of data from satellite measurements and reanalysis source [64]. However, there is no shortage of such studies, although many authors aim to compare different sources of wind speed data and compare it with terrestrial ones. Overall, a SWOT analysis helps the identification of strengths and weaknesses of the strategy to achieve its goals, pinpointing opportunities to reduce the impact on the goals [27]. Tables 1 and 2 summarize a survey of the factors belonging to different SWOT compounds. Information was disaggregated into two subcategories to highlight specific topics for consideration and discussion. The next paragraphs are devoted to present sectorial specificities.

**Table 1.** SWOT (Strengths, Weaknesses, Opportunities and Threats) analysis of satellite potential.

| Forces | Internal | External |
|---|---|---|
| Satellite Potential | Strengths<br><br>ESA and Italian Space Agency (ISA). Open access and unlimited policy support. Wind atlas mapping (wind industry). | Opportunities<br>Save time and money. Knowledge transfer from/to research centers and universities. A better understanding of forgotten areas of whole world (cover existing measurements gaps) |
| | Weaknesses<br>The uniqueness of the available information due to the rotation of the satellite in the Earth's orbit. | Threats<br><br>Specialized knowledge in various interdisciplinary fields. |

**Table 2.** SWOT analysis of potential assessment aspects.

| Forces | Internal | External |
|---|---|---|
| Potential Assessment | Strengths<br>Marine large area observation. Find area with good geographic position and energy potentials (hot spot). | Opportunities<br>Better understanding of sources. Reduce strategic risk. Maximum use of susceptible areas. |
| | Weaknesses<br>Sources estimation in Italy is incomplete. Incomplete measurable parameters. | Threats<br><br>Climate change. |

These categories provide an overview of the factors for using wind speed measurement satellites in areas of interest with different perspectives which could help to better understand the capability of this tool in measuring wind energy. Furthermore, this analysis is based on a review of the literature in two categories in different dimensions and also uses the support tools used in the ODYSSEA project (the platform is available on, http://odysseaplatform.eu/).

## 4. Results

In this section, the results of the SWOT analysis were summarized, considering the two parts of RS potential and potential assessment.

### 4.1. Remote Sensing Techniques

Strengths: Recently, ESA designed, developed, and launched a new family of satellites called Sentinel (includes S1 to S6) as a part of the Copernicus Program. Sentinel-1 (S1, SAR) is able to perform very detailed analysis in this area and would like to thank the different polarization modes: single polarization (vertical-vertical (VV) or horizontal-horizontal (HH) and dual-polarization (VV + VH or

HH + HV) [65]. The VV polarization is very useful for detecting wind speed in ocean and also for understanding the different kind of ocean and sea activities such as fisheries, ship routing, coastal surveillance, offshore installations and exploration. The main reason to select the VV polarization is because of its success in detecting wind speed, since this kind of polarization is sensitive to sea and ocean surface roughness (sea surface water). Images obtained at VV polarization by the SAR satellite are highly sensitive to wind speed variations by means of RMSE which is lower for VV polarization than HH polarization for Sentinel-1B [66]. For C band SAR images analysis, such as Sentinel 1, the C-band model CMOD (C geophysical model function) family (such as CMOD 4, CMOD 5, CMOD 5.n, CMOD 7) and a new model function called C-SARMOD2 can be used. Measuring surface roughness caused by wind is an important feature of the SAR images. SAR satellite capability depends on Doppler information to achieve good resolution in the along-track direction [31].

The Italian Space Agency, along with the Italian Ministry of Defense have developed the COSMO-SkyMed system which is the unique constellation of four radar satellites for earth observation. Those four radar satellites of COSMO-SkyMed system have advanced technology and uses high-resolution radar sensors to observe the earth's day and night, regardless of weather conditions with varying revisit time. Main theme areas are emergency prevention, strategy, scientific and commercial purposes, providing data on a global scale to support a variety of applications among which forest & environment protection, risk management, natural resources exploration, land management, maritime surveillance, defense and security, food and agriculture management. The COSMO-SkyMed satellites have main payloads of X-band, multiresolution and multi-polarization imaging radar, with various resolutions (from 1 to 100 m) over a large access region. Since, it is equipped with a fixed antenna, having electronic steering capabilities that could manage many operative modes for the image acquisition and for internal calibrations. The nominal incidence angles varied between 20° and 59°.

SAR images have great potential for the observation, monitoring and detection of marine sources. It is important to have a wind parameters long-term reference analysis that can cover a large geographical scale. This is more important when many of the observations made from ocean tool measurements are scattered [67]. One of the most important reason to use S1 satellite data and software is because of its free access, supported by unlimited policy with just a sign-up before trying to download the images. However, the data are systematically provided by delivering within an hour of reception for near-real-time (NRT) emergency response, within three hours for NRT priority areas and within 24 h for systematically archived data. Images taken from satellites at different frequencies are used to analyze and map wind parameters at of the seawaters. Recently, the OW field retrieval method has been developed based on satellite data sources and image processing techniques [68]. In many of the Mediterranean coastal, nearshore, and offshore areas, there are no observations of the tall tower for validation. Hence, maritime validation relies on reports of ships and buoys.

Not only SAR satellite imagery can provide OW field data with a long time-series of large and small zones, but also satellite imagery is playing a significant role in offshore observation and research. Even though sea-level wind measurements can be carried out using buoys in different periods with very high resolution, these buoys are usually installed at a distance of 10 to 100 km from the shoreline indicating the possibility of the lack of access at greater distances [69]. Ocean winds recorded from scattering and radiometers have a higher temporal resolution [70]. This higher temporal resolution can be improved by using the cooperation of several satellite data. Due to the increase in the number of observations and time resolution, the accuracy of OW sources estimation can be gradually improved [71].

Furthermore, the ERA-interim reanalysis dataset is a reanalysis project dataset designed and developed by the ECMWF [37]. The ECMWF uses predicted models and data capture techniques, including 4D analysis with a 12 h analysis to describe the atmospheric parameters of the land and oceans such as wind speed, evaporation, surface pressure, surface roughness and surface net solar radiation [72]. In ECMWF dataset, the wind speed at 10 m height, 10 m U wind component, 10 m

wind speed and 10 m V wind component are available since 1979. The intended spatial resolution is characterized by monthly, yearly and four hourly intervals for each day [73].

Even though the vastness and infinity of wind energy in many parts of the world are having good wind power [74], the problems related to the determination of wind measurement accurately making it out the scheme more difficult. The availability of meteorological and floating data information especially in coastal areas, making the situation difficult by lack of access to the complete data set for a study area [75]. On the other hand, good data are available to identify suitable locations through meteorological models and satellite observations. It should be noted that the reanalysis data collects a complete form of data on terrestrial existence, for example, meteorological stations, buoys and cub anemometers, ships and satellite data, which can provide a more accurate display of wind resources on a scale. Such data are regularly monitored with high quality without delay (unlike floating devices: buoys and cub anemometers). In this case, reanalysis data showed the lowest overall error compared with buoys and ship data [76].

Due to the dramatic increase in the use of renewable energy around the world, the need to identify suitable areas for installation and the size of wind farms has increased. In this case, the reanalysis data can be used with a great ability to identify these areas. For example, at different altitudes, it can measure the wind for industrial use of the area, so we called industrial wind. Which can show itself in the view of an accurate wind atlas. Wind atlas contains different wind parameters in different regions, for example, maps, wind speed and direction, time series and frequency distributions. A wind atlas covers the average of important wind parameters at different altitudes for long periods that can come to governments, companies, and academic projects, for example (https://globalwindatlas.info/). An atlas map of offshore wind energy potential that considers all wind parameters can identify potential areas for the nearshore and offshore wind farms installation for later stages [77].

Opportunities: RS data and techniques guarantee a high level of reproducibility because S1 data (SAR) is free and has global coverage. To be more precise and accurate, SAR images with longer time intervals should be used. SAR satellites have several advantages, such as the high spatial resolution, the coverage of large areas, obtained in all-weather conditions, day and night (24 h) [78]. The ability to identify hot spots or focus on ROI makes it an interesting tool for preliminary analysis with different goals and by different target groups. On the other hand, the researcher can obtain data over a long period of time. Many researchers used the reanalysis dataset, which is long term time series of data on wind speed analysis, for network integration studies on wind potential areas [79]. The most important advantage of reanalysis dataset is that they are generally free. This type of information received from the global observation system are made up by different observations tools such as satellites, meteorological stations, and ships to cover a large area [80].

Weaknesses: There are several sensor options for measuring wind at sea and ocean waters using satellite RS, but those satellite and sensors like many other measurement tools, has limitations. For this reason, many researchers have used synthetic winds derived from multimeter scattering, radiometry, and reanalysis data [81,82]. In addition, mostly the assessments were confined to a single or dual scatterometer or using different data sources like reanalysis model, a single scatterometer data is limited to a specific period of time [81]. On one hand, SAR satellite images relative with ocean and sea water surface usually manifests expressions of atmospheric phenomena occurring in the boundary layer can be attributed to the phenomenon [83], such as boundary layer rolls, atmospheric convective cells, atmospheric internal gravity waves, tropical rain cells [84]. Furthermore, SAR satellites have a limit on scanning different areas, which can include one or two scans per week and/or day and are not universal. Much of these satellite data can be accessible with some restrictions of user need (users need a proposal), such as TerraSAR-X, COSMO SkyMed and Radarsat-2. These restrictions also include time constraints with the launching time of satellites. The accuracy of offshore wind assessment using SAR satellite imageries may be affected by different reasons, such as land contamination and lower water depth in the coastline areas [56].

Other factors such as image quality, in-orbit time of the satellite and also hard targets such as oil spills, land, islands, ships, WTGs and WECs are limiting ability to measure wind at the ocean and sea areas from the satellite's imageries. Especially while researchers using the SNAP software, all these hard targets should be masked at the first stage of the processing. For example, in SAR image processing for wind field estimation, the studied areas should be checked out for oil spill and removed from the image because of decreasing the water roughness of the sea surface [85]. Another important weaknesses, the wind direction (SAR images can be analyzed in SNAP software to achieve wind direction with 180 degree ambiguity) obtained from the SAR satellites cannot be verified because the available data will not really show the wind field in the coastal and open seas. This is the main reason for selecting local data (in situ data) in many studies. Another limitation of wind speed estimation based on satellite data are that they measure wind speed data at 10 m above sea surface water. By using this type of data to obtain information about the wind sources at a height of 100 hub meters, it is necessary to implement theoretical models using surface roughness coefficient [76].

Threats: Interdisciplinary knowledge is needed to work with RS methods as this involves obtaining and analyzing satellite data. Then, researchers need to be familiar with various software applications to properly analyze satellite data. It is known that satellites are highly capable of observing the Earth, since they have applications in many different fields. Hence, studying the specific application mainly depends on the software knowledge of researchers.

## 4.2. Potential Assessment

Strengths: Long-term data collected from meteorological stations have generally been used to investigate the wind energy potential for an area. Given the limitations of measuring and installing marine buoys in offshore areas, much research needs to be conducted to identify the areas that are not considered. In this regard, satellite data are used in many aspects, for example, marine engineering, numerical model, oceanographic, wind speed and wave height [86]. In this area, many studies have already been conducted by various researchers on the estimation of marine wind energy on spatial scattering data greater than 12.5, 25 and 50 km resolutions [87,88]. This wind maps can be improved either by one-kilometer grid resolution [89] from Sentinel 1 data. Majidi Nezhad et al. [57] explained a new method for assessing, reporting, and mapping the wind energy potential of sea areas using Sentinel 1 imageries in the Sicilian island of the Mediterranean Sea. First, they identified the hot spots for wind turbine and wind farm installation in large marine areas and accordingly estimated the average wind potential in small areas around the islands. Sentinel 1 satellite images have been analyzed by using the SNAP software and then the wind parameters mapped in the GIS software. At the end of the SAR imagery analysis and mapping, the mean wind regime was extrapolated using the ROI tool and was used as an input data to train and test the proposed forecast model [68].

Figure 6 shows the mean wind speed (MWS) per (m/s) in two different cases (Sicilian and Sardinia islands) using ECMWF reanalysis dataset (between 1979 to 2019). Mediterranean Sea islands usually have higher average wind energy compared to the Mediterranean Sea region in the mainland (Figure 7). The case studies (Sicily, Sardinia) showed wind speeds are higher than the European continental shores. The main reason for this situation is the lower surface roughness (natural barriers) of the ocean surface compared to the ground land.

According to various studies conducted in the Mediterranean Sea, the highest values of wind speed can be observed in the Aegean Sea, the Gulf of Lyon and the Alboran Sea with wind speeds of more than eight meters per second. In the Mediterranean Sea, some areas such as the Aegean Sea, the Strait of Gibraltar, the Gulf of Lyon and the area between Sicily and the coast of Tunisia also have wind speeds of about seven meters per second. The highest increase in wind speed in these areas is observed in spring and summer. This also causes seasonal fluctuations in these regions. Furthermore, during the winter, a significant reduction in wind energy potential is observed in the central parts of the Mediterranean Sea, as in the coasts of Libya and Egypt [46].

These areas, given their good potential, can be called hot spot areas that can be focused on installing power converters. Majority of the Mediterranean islands are characterized by a strong economy from tourism, and these hot spots areas were used to generate renewable energy for the self-sufficiency of the islands also during the seasons when the energy demand significantly increases.

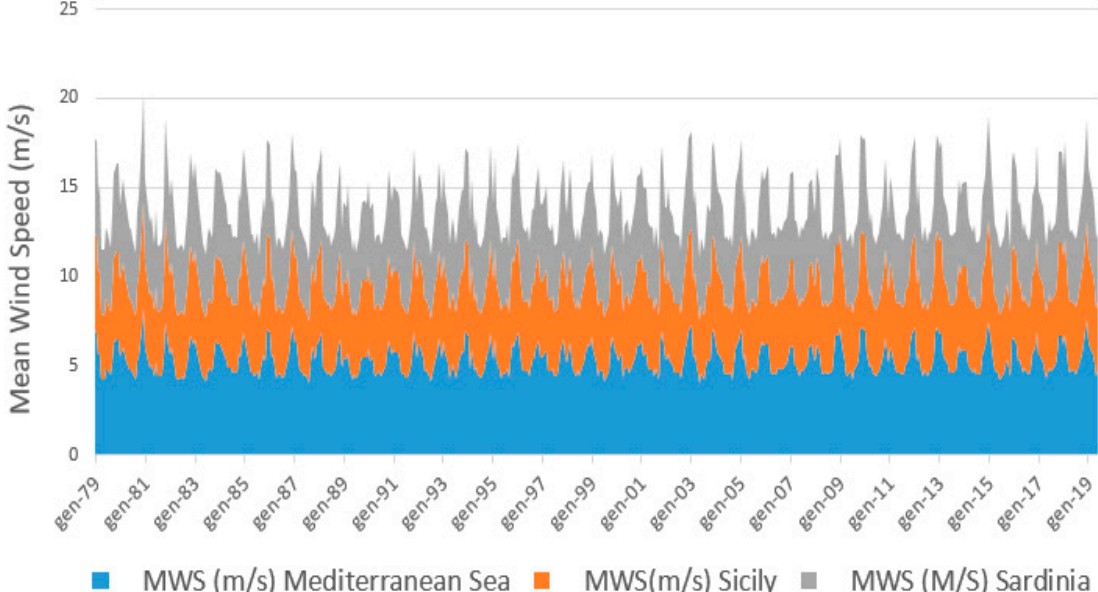

**Figure 6.** Mean wind speed (MWS) per (m/s) with the 10 m height in two different cases and Mediterranean Sea using European Centre for Medium-Range Weather Forecasts (ECMWF) reanalysis dataset (between 1979 to 2019).

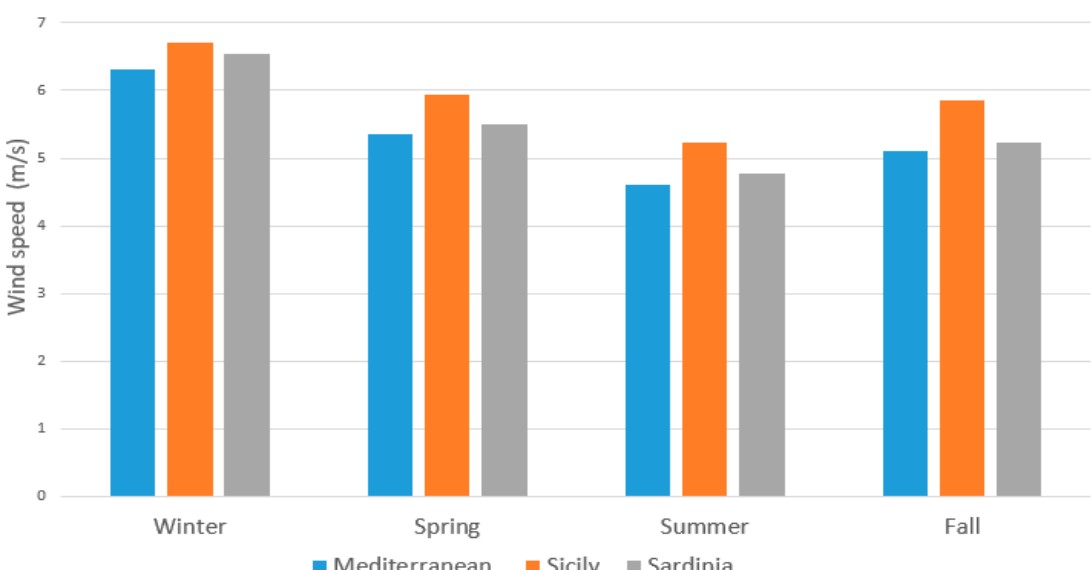

**Figure 7.** Seasonality wind speed (m/s) for case studies.

Opportunities: Since none of the wind farms were shut down to date, it can be treated as a new industry and also as a starting point, given its partnerships with the oil and gas industry [89]. One of the main benefits of OW energy are higher wind energy sources and potential electricity generation compared to land-based wind farms due to the higher wind speeds in the seas and oceans. It also has limited areas to explore given the greater scope for installing marine farms [90]. Sites with a short distance from the coastline are the most attractive location for wind farms installation, since

installation costs increases with increase in the distance from the shore, mainly due to two related factors: water depth and cable cost [67]. Wind energy is the most popular type of renewable energy and its popularity is directly related to harmlessness. It is economical and environment friendly and can play an important role in reducing $CO_2$, SOx and NOx [10]. Using satellites to monitor wind energy and analyze changes in offshore wind energy is crucial for installing wind farms and for observing large areas. Such analysis often requires case study data that are particularly important for developing accurate business models for working on new offshore wind farms, considering that their number is expected to dramatically increase in the following years.

Weaknesses: The potential of Italy's OW energy source is outstanding in many places. The Italian government set a goal of 950 MW for OW by 2030 in its national energy and climate plan presented to the European Commission. Therefore, there are good strengths and opportunities for the Navy in Italy. However, the development of the Italian naval force is still at an early stage and still faces many challenges and threats. Government can set the strongest possible priorities for the future development of maritime zones in the best possible way with the clear and coherent maritime policies they adopt. Such policies should reflect specific goals aimed at establishing hydro power as a means of achieving some reduction in carbon dioxide and clean energy production [7]. Many offshore wind projects in Italy have been cancelled mainly due to lack of funding or opposition from local authorities and the small distances from shore. Moreover, many other projects are in a dormant state during the initial planning phase. OW farms continue to suffer from weaknesses, as prices within the OW power network were uncertain and so investment risk was higher. On the other hand, the development of OW power involved several parts such as planning conflicts between various departments and dealing with problems [91]. Large-scale wind turbines have become a mainstay of technology development in the world. Many countries declared that they were producing more wind turbines and the power plant was concentrated at eight megawatts to four megawatts [91].

Threats: Wind energy potential varies with the cube of wind speed, making minor changes to wind circulation patterns and severity will have a profound impact on future wind power generation. This indicates the sensitivity of wind energy which is related to climate change and future changes. Moreover, wind flow characteristics can also severely affect the ability of wind energy potential usage [92]. For example, a change in the year-to-year wind velocity impacts on electricity generation capability, and the higher the volatility, the more variable the power generation may be, which can cause problems in demand on the electricity grid. On the other hand, there is a decrease in the profitability of wind farms. Considering this parameter, this is an important issue for the economic feasibility of a wind farm having lifetime of about 20–25 years [92]. Climate change also affects other factors, such as water depth and distance at the coastal areas.

The high cost of investment is one of the most important reasons limiting the development of OW farms construction. OW energy sector includes many professional expertise and institutions: the maritime department, the environmental protection agency, the fisheries and military departments. The coordination between them could be very difficult, especially during the planning and approval processes [91].

## 5. Discussion

The purpose of this research is to identify and investigate the potential of satellite and reanalysis data for finding out the suitable areas for the installation of wind farms. A SWOT analysis was carried out to examine the various points of this study, i.e., remote sensing techniques and potential assessment which can help in better usages of new technologies such as RS for the development in the field of renewable energy by applying a suitable approach. Since there is an increasing attention in the widespread usage of clean technology in various societies, new measuring instruments need to be replaced by traditional instruments. For example, with increase in the height of wind turbines, the height of wind turbine towers also increases, which requires much money and time (which, in

turn, takes more than one year to receive data from in situ measurements). With this approach, a preliminary study is required to find out the areas suitable for wind farms.

There is a possibility to use satellite technology for measuring various parameters mainly wind speed in different parts of the world. Today, the use of satellite data can be observed with a view of wind atlas, which is very suitable for the development of new wind farms. On the other hand, it helps governments, companies, and universities to focus more on forgotten areas for expanding the use of RESs. The factors influencing the development of renewable energy are not limited to their measurement technology, so factors such as social, economic, legal, technical, and environmental aspects must also be considered. In such cases, SWOT analysis would be helpful in selecting several factors and eliminating some other factors to limit research. In other point of view, it should not be forgotten that there is a possibility of ignoring some factor without the proper knowledge since this comes into other discipline of research. To avoid this problem, is important to work in collaboration with researchers from different fields.

Considering various aspects of the renewable energy assessment using satellite technology, the main issues to be considered are the obtainment of satellite data from space agencies or from the collaboration with universities with strong aerospace field. Moreover, the emergence of interdisciplinary disciplines and knowledge transfer between them is of importance. Efforts in increasing transparency, communication, and participatory policies may, in-turn, increase the use of each of these sectors in creating coherence between multiple stakeholders.

A better understanding of satellite measurement technology will help to use the potential of this tool in the economic field of projects in a better way, challenging the economic sector by reducing investment costs. By using satellite data, priority list can be created for the construction of new wind farms, taking factors into account such as environmental protected areas, tourist activity, maritime transport, fishing, etc. which could help in simplifying the decision of any of the desired sections.

Furthermore, an important issue to be discussed is that a wrong assessment of wind speed can lead to the failure of a wind farm project. Consequentially, due to high cost in using traditional methods, a preliminary study of the area is necessary. This initial study allows to obtain the rate of change of wind parameters in long- and short-term scenarios, starting from satellite data, facilitating their prediction in the future.

One of the major problems in most offshore areas is the lack of high-precision information. Due to the high cost of installing the device or the lack of calibration, these devices are sometimes less accurate, which, in turn, creates gaps in geographic and temporal data in the basin. In this regard, these gaps can be eliminated with the implementation of new methods using satellite data and a method to prioritize these gaps can be consequentially developed.

## 6. Conclusions

Trying to evaluate the technical suitability of areas for wind farms installation requires a clear conceptual framework that demands examining the potential of appropriate tools and installation locations. To achieve this goal, it is necessary to identify the technical potential that can be defined as a suitable framework for evaluating energy potential. The purpose of this article is to identify and study factors that could halt or encourage the development of satellite remote sensing with a focus on OW detection using a SWOT analysis, which is an appropriate tool for comprehensive research. The main limitation of SWOT analysis is its autonomy, which depends on the analyst's choice of factors to be considered.

In addition, another limitation of this type of analysis relates to the loss of information or compensation during processes when collecting information. Due to the development of satellite remote sensing techniques related to the wind field estimation from sea surface water, SWOT analysis is important to better-understand the importance of satellite data in offshore region. Obtained results pinpointed that there are currently several options for measuring wind speeds in remote areas such as satellite remote sensing, water forecasting models and numeric models, identifying limitations and

strengths of each set of data. Undoubtedly, with the advent of marine energy-based technologies, good planning based on the factors driving new technologies and the potential of these approaches is an essential step.

First, according to the studies of satellites ability to measure marine renewables data, regional measurements and simulation results with the long-term implementation of numeric models with high spatial resolution, possible effects of wind farm facilities in the oceanography installation site, accurate information on protected areas and marine habitats were understood. Second—according to the identified pros and cons of using these techniques—it supports corporate investment in the development of new indigenous technologies and the harmonization of existing foreign technologies for further use at the installation site. In addition, there will be a support of government for OW farms as a logical solution to support energy self-sufficiency—especially in remote areas of the mother country such as the islands and also a significant step to reduce environmental impacts, promoting OW energy that has evolved in Europe (such as the northern regions) to achieve commercial and financial development. Lastly, since these two dimensions are interconnected, this research underlined that there is a need to clearly understand the framework of equipment and its working principles.

The technology of using satellites to identify suitable areas for setting up wind farms is very promising and requires good planning. This proper planning is possible only with full knowledge of satellite capabilities and different methods for measuring wind sources. For this purpose, it is very useful to have interdisciplinary knowledge and exchange of updated information between different sectors of the development such as universities and companies. For complete understanding, few more factors, such as economic, social, environmental, legal and technological issues for setting up offshore wind farms, must be considered to achieve an appropriate framework. Since all the mentioned steps in the latter stage are interconnected and have a direct impact on each other, so it is important to design a comprehensive path to simultaneously coordinate the study with all effective factors.

**Author Contributions:** Conceptualization, M.M.N., R.U.S. and D.A.G.; methodology, M.M.N., R.U.S. and D.A.G.; investigation, M.M.N., R.U.S., A.H., A.R., N.A. and D.A.G.; resources, M.M.N., R.U.S., A.H., A.R., N.A. and D.A.G.; data curation, M.M.N., R.U.S., A.H., A.R., N.A. and D.A.G.; writing—original draft preparation, M.M.N., R.U.S. and D.A.G.; supervision, D.A.G. All authors have read and agreed to the published version of the manuscript.

**Funding:** This research was funded by European Union's Horizon 2020 research and innovation program under grant agreement No 727277 within the project ODYSSEA "Operating a network of integrated observatory systems in the Mediterranean sea".

**Acknowledgments:** This research was carried out within ODYSSEA project that received funding from the European Union's Horizon 2020 research and innovation program under grant agreement No 727277.

**Conflicts of Interest:** The authors declare no conflict of interest.

## Abbreviations

| | |
|---|---|
| RS | Remote sensing |
| SWOT | Strengths, weaknesses, opportunities and threats |
| RESs | Renewable energy sources |
| MREs | Marine renewable energies |
| OW | Offshore wind |
| WEGs | Wind turbine generators |
| WECs | Wave energy converters |
| SAR | Synthetic aperture radar |
| ECMWF | European center for medium-range weather forecasts |
| ESA | European space agency |
| MWS | Mean wind speed |
| WPD | Wind energy density |
| SWH | Significant wave height |
| SCS | Sea current speed |
| SNAP software | Sentinel application platform |
| ENVI software | Environment for visualizing images |
| ROI | Region of interest |
| GIS | Geographic information system |
| VV | Vertical vertical |
| CMOD | C geophysical model function |

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
