# Peer review of "A SWOT Analysis for Offshore Wind Energy Assessment Using Remote-Sensing Potential"

_applsci, doi:10.3390/app10186398_

Round 1

Reviewer 1 Report

The article suggests a strong need for a SWOT analysis in wind energy assessment, and provides a Mediterranean example in the study referring to satellite uses and reanalysis data, in place of not having instruments in situ for wind measurements. I do not have any arguments with what is said, but would, as a climatologist, have liked to have seen some hard data from the two media in comparison to real measured data, but maybe there are studies that have done that. The concept of SWOT seems to me to be a given in addressing the wind energy problems everywhere.

Author Response

Thanks for your comment. We totally agree with you but researchers are facing problems in measuring wind parameters in large unknown areas around the world. Various studies have been compared with the error rate of in-situ measurement tools (such as, Buoys and cub anemometers) with satellite data and reanalysis dataset. Due to the lack of measuring devices in these areas, there is another open access (free policy support) solution for measuring environmental parameters. But it can be examined with recommended satellite methods which is the purpose of this study. SWOT analysis helps to better understand the use of this technology.

We better clarified it in the paper now.

Reviewer 2 Report

The manuscript is intersting and good written. It fits the scope of the Journal and relates to the important topic of wind speed/power determination potential. Moreover, it helps to obtain the data/information about the regions/hot spots, also in the areas without the traditional systems supported by wind masts. However, there some issues that should be better explained or discussed before acceptance for publication.

General:

  • In the abstract the main results achieved are missing
  • Abbreviation list should be added before the refferences list
  • Table 2 describing the SWOT points of the satelilte potential is not developed satisfactory. There should be more arguments. For instance, a strong point is the ability to map areas that are inaccessible, too distant or difficult for the application of traditional systems (wind masts). The authors must consider the SWOT analysis in a wider range.
  • As the authors present some results form measurements, it would be good to compare the results obtained from traditional tools with the RS,
  • the authors should better discuss the validation process/posibilities.
  • the limitation of RS vs. traditional systems should be better puzzled in terms of OW farms restrictions.

Detailed remarks are below:

Line 49 - "Blue Energy" definition should be added and related refference

Line 50 - sea current are exluded from MRE?

Lines 61-75: there are more crucial factors/limitations influencing OW appliacation such as: sea routes for marine transportation and ships, flying routes for migrating birds, fishing areas, corrosion/erosion problems etc. Many aspects have been overlooked. Please, be more careful and decribe other factors related to OW development.

Line 76 - Please define "OW" before abbraviation.

Line 76: "Globally, OW installed power is 23.1 GW (2018), and 21.1 GW are already" - It is a mistake, as suggest that outside Europe there is no OW. There are many OW farms in India, China etc. Please, make some corrections as a sentence at the present form leads to confusion. "Globally" includes Europe, so the sentence is unfortunate. Rewrite the sentence to pass a message more clear.

Line 89-94: It should be pointed out that the results from sattelites iclude also the "measurements" from minimum one year. Otherwise, the whole picture of the wind potential is impossible. It shlould be better explained. Moreover, the measurement height of the wind speed distribution should correspond to the nacelle location (the best option) - authors should be more precise.

Please, add also some refferences to the literature, where the principles of the operation of LIDAR and SODAR are explained.

Line 95: Do not use abbreviations before it is expalined - SWOT

Lines 158-159: "ms-1"- is it meter per second?, if YES, the unit is written not properly, the multiplier is missing.

Line 208, 232 etc. - the units m/s, kW/s are not written properly

Line 323 etc.: remove double-spacing

Line 358: Odyssea project - the refference is missing.

Line 373 etc. - do not use abbreviation before its definition 

Figure 6 and 7 - the quality is poor, the numbers are too low, improve the readability of these figures, description of axes are too small etc.

Author Response

The manuscript is interesting and good written. It fits the scope of the Journal and relates to the important topic of wind speed/power determination potential. Moreover, it helps to obtain the data/information about the regions/hot spots, also in the areas without the traditional systems supported by wind masts. However, there some issues that should be better explained or discussed before acceptance for publication.

Thanks for your comments and positive view.

General:

  • In the abstract the main results achieved are missing

The abstract has been thoroughly modified.

  • Abbreviation list should be added before the refferences list

We added this part.

RS

Remote Sensing

SWOT

Strengths, Weaknesses, Opportunities, and Threats

RESs

Renewable Energy Sources

MREs

Marine Renewable Energies

OW

Offshore wind

WEGs

Wind Turbine Generators

WECs

Wave Energy Converters

SAR

Synthetic Aperture Radar

ECMWF

European Centre for Medium-Range Weather Forecasts

ESA

European Space Agency

MWS

Mean Wind Speed

WPD

Wind Energy Density

SWH

Significant Wave Height

SCS

Sea Current Speed

SNAP software

Sentinel Application Platform

ENVI software

Environment for Visualizing Images

ROI

Region of Interest

GIS

Geographic information system

VV

Vertical Vertical

CMOD

C Geophysical model function

  • Table 2 describing the SWOT points of the satelilte potential is not developed satisfactory. There should be more arguments. For instance, a strong point is the ability to map areas that are inaccessible, too distant or difficult for the application of traditional systems (wind masts). The authors must consider the SWOT analysis in a wider range.

We improved this part and explained

  • As the authors present some results form measurements, it would be good to compare the results obtained from traditional tools with the RS,

“It should be noted that the reanalysis data collects a complete form of data on terrestrial existence, like, meteorological stations, buoys and cub anemometers, ships and satellite data, which can provide a more accurate display of wind resources on a scale. Such data is regularly monitored to maintain high quality without delay (unlike floating devices: buoys and cub anemometers). In this case, reanalysis data showed the lowest overall error compared with buoys and ship data [76].

The purpose of this article is to identify and study factors that could halt or encourage the development of satellite remote sensing with a focus on OW detection using a SWOT analysis, which is an appropriate tool for comprehensive research. In addition, another limitation of this type of analysis relates to the loss of information or compensation during processes when collecting information. Due to the development of satellite remote sensing techniques related to the wind field estimation from sea surface water, to better understand the importance of satellite data in offshore region, SWOT analysis is important.

  • the authors should better discuss the validation process/posibilities.

We improved this part, thanks for your comment. Done

  • the limitation of RS vs. traditional systems should be better puzzled in terms of OW farms restrictions.

We improved this part, thanks for your comment. Done

Detailed remarks are below:

Line 49 - "Blue Energy" definition should be added and related reference

Done.

Line 50 - sea current are exluded from MRE?

Done.

Lines 61-75: there are more crucial factors/limitations influencing OW appliacation such as: sea routes for marine transportation and ships, flying routes for migrating birds, fishing areas, corrosion/erosion problems etc. Many aspects have been overlooked. Please, be more careful and decribe other factors related to OW development.

We added. “. Furthermore, there are more crucial factors/limitations influencing Offshore Wind (OW) applications such as, ship sea routes for marine transportations, migrating birds, economic activities (e.g. fisheries areas), environmental constraints (marine protected areas) and landscaping view”

Line 76 - Please define "OW" before abbraviation.

We added in line 72 (first time).

Line 76: "Globally, OW installed power is 23.1 GW (2018), and 21.1 GW are already" - It is a mistake, as suggest that outside Europe there is no OW. There are many OW farms in India, China etc. Please, make some corrections as a sentence at the present form leads to confusion. "Globally" includes Europe, so the sentence is unfortunate. Rewrite the sentence to pass a message more clear.

We rewrote that sentence try to be clearer.

In general, the installation capacity of wind farms in 2019 for European countries (ECs) 21.1 GW, in 2020, with an expectation of 25 GWh totally in OW energy production, and it is expected that by 2030, it will reach 70 GW of offshore installation capacity [15][16]. In total, ECs have 4,149 grid-connected wind turbines and 81 offshore wind farms installed, which are used only in 10 countries of northern Europe. According to 2017 data, about 50% of offshore wind farms in continental Europe is installed in the UK (53% of the net 3.15 GWh of installed capacity in Europe). By 2024, Europe's total installed capacity is expected to reach 29.8 GW, expanding at an annual growth rate of 12%.

Line 89-94: It should be pointed out that the results from sattelites include also the "measurements" from minimum one year. Otherwise, the whole picture of the wind potential is impossible. It should be better explained. Moreover, the measurement height of the wind speed distribution should correspond to the nacelle location (the best option) - authors should be more precise.

Satellites are the only tool that can measure the average, minimum and maximum wind speed in the study areas (hot spot areas) in the shortest possible time over a long period of time. It should be noted that the reasons for the popularity of this data in the research and academic communities can be mentioned as follows, i) This data is generally available for free (open access), ii) They can cover a period of more than 40 years. Due to the fact that it is not possible to install ground wind gauges (such as, cub anemometers) in different areas, due to its high cost, satellites are the only device that can cover areas for more than a year, which is an important factor in assessing wind resources [21]. Satellite devices for observing, reporting and evaluating RES potential have led to a revolution in the installation of energy converters in new locations that have not previously been considered. In addition, due to the increase in surveys to identify industrial wind at an altitude of more than 100 meters, various atlases have come out a san outcome. It takes long time to install anemometers on-site to measure industrial winds at higher altitudes, which can be done with satellite data. However, in order to point out the best options and strategies, all aspects of strengths, weaknesses, opportunities and threats must be considered.

Please, add also some refferences to the literature, where the principles of the operation of LIDAR and SODAR are explained.

Goit et al [22], explained the though reconstruction from LiDAR-measured radial wind speed to wind speed vector is a challenge, LiDAR-based wind speed measurements are undergoing a significant increase in interest for wind energy application. Here, the study employed the scanning of Doppler LiDAR for assessment and comparison.  Firstly, the evaluation of the effect of Carrier-to-Noise-Ratio (CNR) and data available on the quality of scanning LiDAR measurements was done.  Then, it was proposed to reconstruct the wind fields from Plan-Position Indicator (PPI) and Range Height Indicator (RHI) scans of LiDAR-measured line of sight velocities. It was observed that an internal boundary layer with strong shear could be developed from the coastline. Lastly, PPI scan was involved to measure the flow field around a wind turbine and validate wake models. Chaurasiya et al [23], to increase the confidence of RS technique to compute Weibull parameters at higher heights for assessment of wind energy resource. It is known that RS techniques are gaining attention worldwide for the comprehensive assessment of wind source in flat, complex, and mountainous terrain. The 10 min average time series wind speed data for the period of one month (in September 2014) were recorded simultaneously at 80 m and 100 m using the cup anemometer installed in the proximity of 120 m meteorological mast, second wind triton SODAR (Sound Detection and Ranging) and continuous wave wind LiDAR (Light Detection and Ranging). Nine different methods were implemented to analyse and obtain Weibull parameters on the data obtained from the measurements. Totally, there’s an expectation that the outcome of this study could encourage the deployment of remote sensing techniques at Indian sites.

 Line 95: Do not use abbreviations before it is explained – SWOT

Done.

Lines 158-159: "ms-1"- is it meter per second?, if YES, the unit is written not properly, the multiplier is missing.

Done.

Line 208, 232 etc. - the units m/s, kW/s are not written properly

Done.

Line 323 etc.: remove double-spacing

Done.

Line 358: Odyssea project - the refference is missing.

Done.

Line 373 etc. - do not use abbreviation before its definition 

Done.

Figure 6 and 7 - the quality is poor, the numbers are too low, improve the readability of these figures, description of axes are too small etc.

Done. We improved the readability of these figures.

Reviewer 3 Report

The topic in the manuscript “A SWOT Analysis for Offshore Wind Energy Assessment using Remote Sensing Potential” could be interesting. However, in its current state the manuscript is very disorganized, lacking a proper narrative and structure.

It is not clear if the manuscript is a research paper or a literature review. It is not clear which results are new and original from the authors, which results correspond to research previously published by the authors and which results are from other researchers. For instance, it is not clear if figures 2 – 5 are results or literature review from reference [50]. It is not clear if results in Figure 6 were already published by the authors (reference [62]) or are original for this paper. The authors need to clarify if this is a literature review or a research paper.

The ideas are presented in a disorganized fashion, with no clear narrative or order. I would recommend including a flow diagram containing the main ideas discussed in the manuscript in logical order to give readers a clear narrative path to follow. Topics on renewable energy, offshore renewables, offshore wind energy, remote sensing, applications and results are scattered all over the manuscript. Each topic needs to be explored individually and when its analysis is complete move to the next topic. The manuscript If the topic discussion is left incomplete in one section and it is restarted in the middle of another topic,  it becomes very difficult to follow the narrative.

The methods and materials for this paper are disorganized and placed in other sections of the manuscript. Most of the material in the current methods and materials section should be in the introduction, such as the description of the area of study, offshore wind descriptions and wind speed maps from previous literature. This section needs to include organized information about the satellite remote sensing platforms and instruments being considered for this research, including temporal and spatial resolution, instruments on board the platform, capabilities of instruments, swath and potential costs for accessing data. The use of a table containing this information would be useful. Table 1 contains information that is unexplained to the reader in connection to the topics discussed in the manuscript. For instance, why it is important to indicate the Launcher or Inclination in the data? How are the thematic areas mentioned connect with this research?  

Some statements need to be further clarified because can lead to confusion:

Line 123 “high replicability of the adopted methods will lead to significant time savings in identifying potential areas and reducing measurement costs.” How can this be achieved.

Line 140 “A family of satellites with which our land, ocean and atmosphere are observed from space and then provides us the data for free of charge to maximum use of satellites with 24/7 and 365 days a year of availability is called the sentinel family” Is all data free of charge? What does it mean with 24/7 and 365 days a year of availability considered that satellites have orbital periods with specific revisit times per area.

Line 386 “It is at the forefront of technology and uses high-resolution radar sensors to observe the earth’s day and night, regardless of weather conditions.”. What is the meaning of forefront in this context? This statement may lead to inaccurate assumptions from readers regarding continuous observation by satellites. What is the revisit time of the satellites under consideration? It the time of the revisit standard (always same time of day or changes depending on revisit)

Line 400 “One of the most important reason to use S1 satellite data and software is because of its free access, supported by unlimited policy with just a sign-up before trying to download the images” Is all data free? Is the data restricted by nationality? Should export controls be mentioned?

The sections indicating Strengths, Weaknesses and Treats are scattered all over the manuscript with no discernible pattern or narrative integration. I recommend creating a table, which summarizes these strengths, weaknesses and treats, and afterwards explain them in detail.

The English writing and grammar should be completely reviewed. There are several very long statements (4-5 lines). This is not adequate.

I believe the manuscript should be completely rewritten, clarifying if it is a literature review or research paper. It should organize the ideas and topics discussed in a clear narrative, with logical order, integrating each topic in its corresponding section.

Author Response

The topic in the manuscript “A SWOT Analysis for Offshore Wind Energy Assessment using Remote Sensing Potential” could be interesting. However, in its current state the manuscript is very disorganized, lacking a proper narrative and structure.

Thanks for your comment.

It is not clear if the manuscript is a research paper or a literature review. It is not clear which results are new and original from the authors, which results correspond to research previously published by the authors and which results are from other researchers. For instance, it is not clear if figures 2 – 5 are results or literature review from reference [50]. It is not clear if results in Figure 6 were already published by the authors (reference [62]) or are original for this paper. The authors need to clarify if this is a literature review or a research paper.

Thanks for your comment. This is a research paper that was compiled using satellite and reanalysis data. Moreover a SWOT analysis on those tools for a wind farm installation has been developed as in many research papers. This study can help to better understand the capabilities of new tools in select suitable areas.

All satellite images are original and part of the results of this research. We modified the paper as suggested.

The ideas are presented in a disorganized fashion, with no clear narrative or order. I would recommend including a flow diagram containing the main ideas discussed in the manuscript in logical order to give readers a clear narrative path to follow. Topics on renewable energy, offshore renewables, offshore wind energy, remote sensing, applications and results are scattered all over the manuscript. Each topic needs to be explored individually and when its analysis is complete move to the next topic. The manuscript If the topic discussion is left incomplete in one section and it is restarted in the middle of another topic, it becomes very difficult to follow the narrative.

All the contexts of this article are very clearly shown in Tables 1 and 2. Each of which clearly limits the sections under study. Given that this is an interdisciplinary article, a good understanding of several disciplines is needed to better understand it.

The methods and materials for this paper are disorganized and placed in other sections of the manuscript. Most of the material in the current methods and materials section should be in the introduction, such as the description of the area of study, offshore wind descriptions and wind speed maps from previous literature. This section needs to include organized information about the satellite remote sensing platforms and instruments being considered for this research, including temporal and spatial resolution, instruments on board the platform, capabilities of instruments, swath and potential costs for accessing data. The use of a table containing this information would be useful. Table 1 contains information that is unexplained to the reader in connection to the topics discussed in the manuscript. For instance, why it is important to indicate the Launcher or Inclination in the data? How are the thematic areas mentioned connect with this research?  

The authors have tried to give the reader a clearer view by first reviewing previous studies (Literature review PART), to be able to constrain the framework to understand SWOT analysis. We modified the paper as required.

Some statements need to be further clarified because can lead to confusion:

Line 123 “high replicability of the adopted methods will lead to significant time savings in identifying potential areas and reducing measurement costs.” How can this be achieved.

Thanks for your comment. We preferred to delete this sentence.

Line 140 “A family of satellites with which our land, ocean and atmosphere are observed from space and then provides us the data for free of charge to maximum use of satellites with 24/7 and 365 days a year of availability is called the sentinel family” Is all data free of charge? What does it mean with 24/7 and 365 days a year of availability considered that satellites have orbital periods with specific revisit times per area.

Sentinel family which is a group of satellites orbiting around the earth with varying revisiting time for observing land, ocean and atmosphere from space and then for providing us the data free of charge anytime around the year (24/7 and 365 days). We modified the paper as required.

Line 386 “It is at the forefront of technology and uses high-resolution radar sensors to observe the earth’s day and night, regardless of weather conditions.”. What is the meaning of forefront in this context? This statement may lead to inaccurate assumptions from readers regarding continuous observation by satellites. What is the revisit time of the satellites under consideration? It the time of the revisit standard (always same time of day or changes depending on revisit)

Just meant to say, four radar satellites of COSMO-SkyMed system have advanced technology and uses high-resolution radar sensors to observe the earth’s day and night, regardless of weather conditions with varying revisit time. We modified the paper as required.

Line 400 “One of the most important reason to use S1 satellite data and software is because of its free access, supported by unlimited policy with just a sign-up before trying to download the images” Is all data free? Is the data restricted by nationality? Should export controls be mentioned?

Yes, all the data is free for general public, scientific and commercial users but provided systematically by delivering within an hour for Near Real-Time (NRT) emergency response, within three hours for NRT priority areas and within 24 hours for systematically archived data.

The sections indicating Strengths, Weaknesses and Treats are scattered all over the manuscript with no discernible pattern or narrative integration. I recommend creating a table, which summarizes these strengths, weaknesses and treats, and afterwards explain them in detail.

Thanks for your comments. We added a Discussion part.

The English writing and grammar should be completely reviewed. There are several very long statements (4-5 lines). This is not adequate.

Done. We improve the english and rewrote many sentences.

I believe the manuscript should be completely rewritten, clarifying if it is a literature review or research paper. It should organize the ideas and topics discussed in a clear narrative, with logical order, integrating each topic in its corresponding section.

Done. We used all the above mentioned comments and suggestions to significantly improve the paper, rewriting many paragraphs as suggested.

Round 2

Reviewer 3 Report

Many of the review comments were addressed by authors.

I would suggest including flow chart indicating the steps for the Strengths, Weaknesses, Opportunities, and Threats (SWOT) analysis applied in this research. This would provide readers with better understanding on the analysis.

Author Response

Many of the review comments were addressed by authors.

The authors thank you for your good comments.

I would suggest including flow chart indicating the steps for the Strengths, Weaknesses, Opportunities, and Threats (SWOT) analysis applied in this research. This would provide readers with better understanding on the analysis.

Many articles have been reviewed and visited in this paper, which shows in SWOT analysis that a table is needed to better show the steps of research. Due to the many factors under study, the use of a flowchart takes a heterogeneous form. Therefore, all the steps understudy are listed in Tables 1 and 2 for readers.